# Modulation of Antioxidant Enzyme Expression of In Vitro Culture-Derived Reticulocytes

**DOI:** 10.3390/antiox13091070

**Published:** 2024-09-02

**Authors:** Hannah D. Langlands, Deborah K. Shoemark, Ashley M. Toye

**Affiliations:** School of Biochemistry, University of Bristol, Biomedical Sciences Building, University Walk, Bristol BS8 1TD, UK

**Keywords:** erythroid, antioxidant enzyme, reactive oxygen species, glutathione peroxidase, peroxiredoxin

## Abstract

The regulation of reactive oxygen species (ROS) in red blood cells (RBCs) is crucial for maintaining functionality and lifespan. Indeed, dysregulated ROS occurs in haematological diseases such as sickle cell disease and β-thalassaemia. In order to combat this, RBCs possess high levels of protective antioxidant enzymes. We aimed to further boost RBC antioxidant capacity by overexpressing peroxiredoxin (Prxs) and glutathione peroxidase (GPxs) enzymes. Multiple antioxidant enzyme cDNAs were individually overexpressed in expanding immortalised erythroblasts using lentivirus, including Prx isoforms 1, 2, and 6 and GPx isoforms 1 and 4. Enhancing Prx protein expression proved straightforward, but GPx overexpression required modifications. For GPx4, these modifications included adding a SECIS element in the 3’UTR, the removal of a mitochondrial-targeting sequence, and removing putative ubiquitination sites. Culture-derived reticulocytes exhibiting enhanced levels of Prx and GPx antioxidant proteins were successfully engineered, demonstrating a novel approach to improve RBC resilience to oxidative stress. Further work is needed to explore the activity of these proteins and their impact on RBC metabolism, but this strategy shows promise for improving RBC function in physiological and pathological contexts and during storage for transfusion. Enhancing the antioxidant capacity of reticulocytes has exciting promise for developing culture-derived RBCs with enhanced resistance to oxidative damage and offers new therapeutic interventions in diseases with elevated oxidative stress.

## 1. Introduction

Due to their role in oxygen transport, red blood cells (RBCs) are particularly susceptible to oxidative stress. This is, in part, due to continual spontaneous autoxidation of haem-bound iron, which results in the production of a superoxide anion and subsequent peroxides [1,2]. In RBCs, around 1–3% of haemoglobin (Hb) undergoes daily oxidation of its ferrous iron, causing oxygen-bound Hb (oxyHb) to form nonfunctional ferric methaemoglobin (metHb) [3]. Ordinarily, cytochrome b_5_ reductase reduces metHb back to oxyHb, and superoxide molecules are dismutated to hydrogen peroxide (H_2_O_2_), which are, in turn, neutralised by an extensive antioxidant system [2,4]. These events are considered to play a determining role in the circulatory lifespan of RBCs, whereby repeated exposure to oxygen promotes autoxidation, and proximity to activated inflammatory cells increases RBC uptake of extracellular reactive oxygen species (ROS) [5,6]. Prolonged and/or high levels of oxidation can overwhelm the otherwise extensive RBC antioxidant protection mechanisms, resulting in denatured haemoglobin (Heinz bodies), membrane protein clustering, impaired deformability, and, under particularly damaging conditions, haemolysis [1,2,7,8]. 

ROS can also negatively impact the bio-preservation of RBCs during storage, with the formation of damaging oxidative lesions reported to significantly increase after just 14 days [9,10,11]. The development of such storage lesions ultimately perturbs the biophysical properties of stored RBCs and, in turn, impedes their final transfusion viability [12,13].

High levels of ROS are also characteristic of the haemoglobinopathies sickle cell disease (SCD) and β-thalassaemia [2,14]. Despite presenting with different clinical manifestations, both conditions are characterised by the generation of highly unstable Hb, excessive autoxidation, and increased levels of ROS, which ultimately compromises RBC integrity [2,14]. There is now substantial evidence implicating oxidative stress in the pathophysiology of β-thalassaemia, whereby elevated levels of ROS are reported to accelerate cell death at the polychromatic stage of erythropoiesis [2,15,16,17,18].

In order to minimise the impact of oxidative stress, RBCs possess a comprehensive suite of antioxidant systems that encompasses both low molecular weight compounds, such as ascorbate (vitamin C) and glutathione (GSH), alongside a series of more complex enzymatic pathways [2,6]. In brief, enzymatically, a copper/zinc isoform of superoxide dismutase (SOD1) mediates the rapid dismutation of the superoxide anion that is generated upon autoxidation [2,6]. The resultant H_2_O_2_ molecule is neutralised by three different enzyme systems acting in parallel to produce oxygen and water. These are glutathione peroxidases (GPx), the cysteine-dependent peroxiredoxin (Prx) family, and catalase [19].

Of the eight GPx enzymes identified in mammals, GPx1 is the most active in RBCs [20,21,22]. GPx1 is a tetramer comprised of four identical subunits that each contain a catalytic tetrad comprised of glutamine, tryptophan, asparagine, and a non-canonical selenocysteine (Sec) residue [23]. This catalytic core, in particular Sec, is essential for the neutralisation of H_2_O_2_ by GPx1 and requires two molecules of GSH to reduce oxidised GPx1 back to an active state [23].

Likewise, GPx4 also utilises GSH but functions to protect cells from oxidative damage by reducing phospholipid hydroperoxides [23]. In other cell types, GPx4 is reported to play a crucial role in maintaining cell membrane integrity and is a key regulator of ferroptosis, a type of cell death dependent on iron that is characterised by lipid peroxidation [24]. In RBCs, GPx4 is emerging as a key player in reticulocyte enucleation and maturation [25]. Loss of GPx4 in mice models resulted in the accumulation of ROS, lipid peroxidation (LPO), ineffective erythropoiesis and consequently, anaemia, a phenotype exacerbated by vitamin E depletion [26,27]. These effects were also associated with impaired reticulocyte maturation, characterised by an accumulation in proerythroblast cells, a dramatic reduction in reticulocytes, and the few resultant reticulocytes produced exhibiting large autophagosomes and defective mitophagy [27]. The knockdown of GPx4 expression is also reported to delay the DMSO-induced erythroid differentiation of mouse erythroleukemia (MEL) cells. However, in mice models that are heterozygous for catalytically silent GPx4, no significant differences in erythropoiesis were observed [28]. A recent in vitro human erythropoiesis study observed reduced levels of enucleation upon GPx4 inhibition to occur independently of oxidative stress, suggesting that in human cells, GPx4 may play an important structural role during enucleation rather than a catalytic antioxidant one [29]. Differences between these observations are proposed to be due to cell type/species differences, whereby auxiliary antioxidant enzymes that catalyse similar reactions are present in one species but not in the other. For example, Prx6 (see below) may help to compensate for the loss of GPx4 and protect against ROS-induced cell death in human cells, but is not present in mice [29,30].

Peroxiredoxins (Prx) are cysteine-dependent antioxidant enzymes that neutralise H_2_O_2_, lipid hydroperoxides, and peroxynitrates [31]. The three Prx isoforms present in human erythrocytes are Prx1, Prx2 (both 2-cysteine), and Prx6 (1-cysteine). Prx1 and Prx2 are found in both the cytoplasm and at the membrane of RBCs [31]. Prx2 is the third most abundant protein in RBCs and is considered an integral component of RBC antioxidant defence [31]. Both Prx1 and Prx2 neutralise ROS by forming intermediary nonfunctional dimers, where the reduction of H_2_O_2_ triggers a disulfide bond between the catalytic cysteine of one Prx and the resolving cysteine of another. A partner enzyme, thioredoxin, completes the catalytic cycle and resolves Prx1 and Prx2 back to their active states [31]. Prx2 is also reported to function as a chaperone protein that can stabilise metHb to prevent ROS-induced aggregation and Heinz body formation [31,32,33,34,35]. Furthermore, Prx2 can bind to the membrane via the band 3 cytoplasmic domain to help protect against LPO and band 3 clustering by competing for metHb and hemichrome binding sites [31,32,33,36,37,38,39]. The role of Prx6 in RBCs is not yet fully understood but is thought to work alongside GPx4 in defence against lipid peroxidation [29,34]. Like GPx4, Prx6 can reduce phospholipid hydroperoxides and exhibits phospholipase A2 (PLA2) and lysophosphatidylcholine acyl transferase (LPCAT) activity, essential for the repair of damaged phospholipids [40].

Several studies have looked to minimise storage-related oxidation of RBCs via the inclusion of extrinsic antioxidants, such as GSH precursors and vitamin E [41,42,43,44,45,46,47,48,49,50,51], or reducing the generation of pro-oxidants by altering oxygen availability under storage conditions [52,53,54,55]. More recently, the suspension of pegylated nanoparticles (PLGA-NP) containing catalase and SOD enzymes in stored blood helped to reduce storage-related oxidation but may introduced other antigenic effects Selenium-based small molecules have also been shown to protect isolated RBCs from H_2_O_2_-induced damage in a Prx2-like manner [56,57]. Interestingly, multiple studies have reported increased antioxidant enzyme expression and/or activity (notably SOD1 and peroxiredoxins [18,58,59]) in blood samples from patients of both SCD and β-thalassaemia, having likely evolved as a compensatory protective mechanism for the elevated levels of ROS associated with these haemoglobinopathies [60]. In support of this, the administration of cell penetrable recombinant Prx2 (PEP1-Prx2) in β-thalassaemia mice models reduced ineffective erythropoiesis, illustrating that the augmentation of endogenous antioxidants may help to improve cell survival during thalassaemic erythropoiesis [61]. Moreover, studies are also beginning to exploit the enhanced antioxidant levels observed in heterozygous β-thalassaemia cells [18,60,62,63], with studies demonstrating these cells to better withstand storage lesions under blood bank conditions and even exhibit improved post-transfusion recovery [64,65]. Finally, GPx4 abundance has also been reported to correlate with protection against RBC storage damage [66], with specific GPx4 polymorphisms shown to modulate levels of oxidative haemolysis during storage [67].

Taken together, these collective observations suggest that enhancing antioxidant enzyme expression could pose a significant benefit to RBCs, with additional antioxidant enzymatic capacity providing the potential to mitigate oxidative stress and improve the lifespan and functionality of RBCs in both physiological and pathological conditions. In previous work, we showed that it is possible to genetically modify reticulocytes by manipulating the immortalised Bristol Erythroid line Adult (BEL-A) cell line to express the enzyme thymidine phosphorylase, and this was replicated in primary stem cell-derived reticulocyte cultures [68]. This therefore led us to explore whether it is possible to enhance the expression levels of antioxidant enzymes in erythroid cells to demonstrate a potential strategy for producing culture-derived RBCs with enhanced resilience to oxidative stress.

## 2. Materials and Methods

### 2.1. Antibodies

The mouse monoclonal antibodies used were as follows: 9E10 (c-MYC) (IBGRL Reagents, Bristol, UK); 7F2 (Prx2) (Bio-Techne Ltd., Oxford, UK); MG1-45 (IgG1) (BioLegend, London, UK); 0411 (GAPDH) (Santa Cruz Biotechnology, Dallas, TX, USA). The rabbit monoclonal antibodies used were as follows: D5G12 (Prx1) (Cell Signaling Technology, Danvers, MA, USA) and D9J9H (Prx6) (Cell Signaling Technology, Danvers, MA, USA). The secondary antibodies used were allophycocyanin (APC)—conjugated monoclonal rat anti-mouse IgG1 RMG1–1 (BioLegend, London, UK), swine anti-rabbit HRP (P0399), or rabbit anti-mouse HRP (P0260) (both Agilent, Santa Clara, CA, USA).

### 2.2. BEL-A Cell Culture

BEL-A (Bristol Erythroid Line—Adult) cells were cultured, as described previously [69]. All cell lines were maintained at a density of 1 × 10^5^ cells mL^−1^ in expansion media: StemSpan Serum-Free Expansion Medium (Stem Cell Technologies, Cambridge, UK) supplemented with 50 ng mL^−1^ stem cell factor (SCF, Miltenyi Biotec Ltd., Surrey, UK), 3 U mL^−1^ erythropoietin (EPO, Bristol Royal Infirmary Pharmacy, Bristol, UK), 1 μM dexamethasone (DEX, Merck Life Science UK Ltd., Dorest, UK), and 1 μg mL^−1^ doxycycline (DOXY, Merck Life Science UK Ltd., Dorest, UK). Full medium changes were conducted every 48 h by pelleting cells via centrifugation at 300× *g* for 5 min. All cell counts were performed using an automated cell counter, and trypan blue staining was used to monitor viability. 

In order to induce differentiation, 1.5 × 10^5^ BEL-A cells mL^−1^ were transferred into primary differentiation medium (Iscove’s Basal Medium (BioChrom; Merck Life Science UK Ltd., Dorest, UK), containing 3% (*v*/*v*) heat-inactivated human AB serum (Merck Life Science UK Ltd., Dorest, UK), 2% (*v*/*v*) foetal calf serum (Hyclone; GE Healthcare, Amersham, UK), 3 U mL^−1^ erythropoietin (EPO, Bristol Royal Infirmary Pharmacy, Bristol, UK), 10 μg mL^−1^ insulin (Merck Life Science UK Ltd., Dorest, UK), 3 U mL^−1^ heparin (Sigma-Aldrich), 500 μg mL^−1^ holo-transferrin (Sanquin Blood Supply, Amsterdam, NL, The Netherlands), 1 U mL^−1^ penicillin/streptomycin, 1 ng mL^−1^ interleukin-3 (IL-3, R&D Systems, Minneapolis, MN, USA), 1 μg mL^−1^ doxycycline (DOXY, Merck Life Science UK Ltd., Dorest, UK), and 10 ng μL^−1^ stem cell factor (Miltenyi Biotec Ltd., Surrey, UK)). After 48 h, the cells were reseeded at 3.5 × 10^5^ cells mL^−1^ in fresh primary differentiation medium. On day 4 of differentiation, the cells were cultured at a density of 5 × 10^5^ cells mL^−1^ in a differentiation medium lacking doxycycline (secondary differentiation medium). On day 6, the cells were given a full medium change and were maintained at 1 × 10^6^ cells mL^−1^. On day 8, the cells were transferred to fresh medium deficient in doxycycline, IL-3, and SCF (tertiary differentiation medium). The cells were then maintained at a density of 1 × 10^6^ cells mL^−1^ in tertiary differentiation medium, with complete medium changes every 48 h until day 12.

### 2.3. Lentiviral Transduction

To generate lentiviral vectors of interest, all open reading frames (ORFs) were synthesised and cloned into the XLG3 vector by GenScript (Oxford, UK). Human coding sequences for each of these enzymes were obtained from the Ensembl database (Prx1: ENST00000319248; Prx2: ENST00000301522; Prx6: ENST00000340385; GPx1: ENST00000419783; GPx4: ENST00000354171; GPx4s: ENST00000611653). All sequences were codon-optimised for mammalian expression. For the Prx constructs, a c-MYC tag was synthesised at the C-terminus. For the GPx constructs, a c-MYC tag was included at the N-terminus. 

HEK293T (human embryonic kidney) cells (Lenti-X 293T, Takara Bio, aint-Germain-en-Laye, France) were cultured in high glucose Dulbecco’s Modified Eagle’s Serum (DMEM) containing 4 mM GlutaMAX-1 (Gibco; Thermo Fisher Scientific, Cambridge, UK) and 10% (*v*/*v*) fetal bovine serum (FBS, Gibco; Thermo Fisher Scientific, Cambridge, UK). Twenty-four h prior to transfection, HEK 293T cells (Lenti-X 293T, Takara Bio, aint-Germain-en-Laye, France) were seeded at 60–80% confluency in 10 mL culture dishes. HEK293T cells were transfected with a calcium phosphate precipitate comprising the lentiviral packaging vectors pMD2.G (5 μg) and psPAX2 (15 μg) and the lentiviral vector of interest (20 μg). The cells were incubated at 37 °C for 18 h before a fresh medium change. Virus particles were harvested 48 h later using a Lenti-X concentrator (Takara Bio, Saint-Germain-en-Laye, France) according to the manufacturer’s protocol and stored at −80 °C. Virus concentrated from the equivalent of half a 10 cm dish of HEK293T cells was added to 2 × 10^5^ expanding BEL-A cells in the presence of 8 μg mL^−1^ polybrene (Merck Life Science UK Ltd., Dorest, UK). After 24 h, the cells were washed three times in PBS and resuspended in fresh media.

### 2.4. Flow Cytometry

A minimum of 0.15 × 10^6^ cells were fixed in flow cytometry fixing solution (PBSAG (PBS + 1 mg mL^−1^ bovine serum albumin [BSA], 2 mg mL^−1^ glucose) supplemented with 1% (*v*/*v*) paraformaldehyde and 0.0075% (*v*/*v*) glutaraldehyde) for 15 min at room temperature prior to permeabilization for 5 min with 0.5% Triton X-100 in PBSAG. The cells were subsequently labelled with the appropriate antibodies and processed using a MACSQuant Analyser 10 Miltenyi Biotec Ltd., Surrey, UK). Where appropriate, live cells were identified and gated for using the viability dye propidium iodine (Miltenyi Biotec Ltd., Surrey, UK), which was added automatically by a MACSquant Analyser. Reticulocytes of differentiated BEL-A cells were identified by gating upon VioBlue-negative populations following live cell labelling with 5 μg mL^−1^ Hoechst 33342 (Sigma-Aldrich) for 30 min at 37 °C in the dark. All data files were analysed using FlowJo, v10.6.1 (FlowJo LLC, Ashland, OR, USA).

### 2.5. SDS-PAGE and Western Blotting

Protein samples were separated by SDS-PAGE using Mini-PROTEN Precast 4–20% gels (BioRad, Hertfordshire, UK) in a BioRad Mini-PROTEAN Tetra Cell system. A total of 1 × 10^6^ cells were harvested at the indicated time points, pelleted, snap-frozen, and stored at −80 °C. For Western blotting, the cells were lysed in 10 μL NP-40 lysis buffer on ice for 10 min and centrifuged at 13,000× *g* for 10 min at 4 °C. The supernatant (clear cell lysate) was then added to an equal volume of SDS-PAGE sample buffer and heated to 95 °C for 1 min. The samples were separated by SDS-PAGE electrophoresis prior to transfer onto a polyvinylidene fluoride membrane (PVDF, Millipore Immobilon-P, Merck Life Science UK Ltd., Dorest, UK). The membranes were blocked in 5% skimmed milk (Marvel) in TBS-T (20 mM Tris, 75 mM NaCl, 0.2% (*w*/*v*) TWEEN-20, pH 7.7) before with the relevant antibodies. The membranes were developed using ECL Western blotting reagents (GE Healthcare, Amersham, UK) and imaging for chemiluminescence using an Amersham Imager 600 machine (GE Healthcare, Amersham, UK).

### 2.6. Lysosome and Proteasome Inhibitors

Inhibition of the ubiquitin-proteasome system (UPS) and lysosomal-mediated degradation pathways was assessed in expanding BEL-A cells. A total of 6.5 × 10^5^ cells mL^−1^ were cultured in a fresh expansion medium, as previously described. Cultures were supplemented with either 5 μM MG132 (Merck Life Science UK Ltd., Dorest, UK), 10 μM leupeptin (Merck Life Science UK Ltd., Dorest, UK), or 10 μM bardoxolone (Generon) for 6 h at 37 °C to inhibit UPS, lysosomal-mediated degradation and chaperone-mediated autophagy (CMA), respectively. DMSO (MG132 and bardoxolone) or ddH_2_O (leupeptin) were used as vehicle controls (VCs) in control samples accordingly.

### 2.7. GPx4 Modelling

The GPx4 coding sequence was examined for putative ubiquitination sites using PhosphoSitePlus [70]. Protein structures were generated using 7L8Q.pdb structure, visualised, and modified in Chimera [71], and the mutated coding sequences were checked using BLAST [72]. Energy minimisation for computations models was performed by generating a box of water containing 150 mM NaCl using GROMACS to ensure the mutations did not disrupt the fold under periodic boundary conditions [73]. Finally, the mutated, energy-minimised protein structure was validated using PROCHECK [74]. 

## 3. Results

### 3.1. Overexpression of Peroxiredoxin Enzymes in BEL-A Cells

To investigate whether the developing red blood cell has additional capacity to express antioxidant proteins, we first explored mammalian peroxiredoxin (Prx) proteins. Polyclonal erythroid cell lines individually overexpressing mammalian Prx1, 2, and 6 cDNA (all c-terminally tagged with c-MYC) were generated by the stable lentiviral transduction of the BEL-A cell line [69]. The resultant cell lines were assessed for overexpression by immunoblotting using commercial antibodies to each enzyme (Figure 1A). The representative Western blots are shown in Figure 1A, illustrating bands present with the molecular weight of both endogenous protein and tagged constructs in the cell lysates of the respective transduced cell lines. Importantly, the abundance of endogenous Prx1, 2, and 6 remained unchanged compared to the untransduced (UT) control, indicating that an overall increase in total Prx enzyme level was achieved for each isoform without any compensatory responses to endogenous protein levels.

In order to quantify the level of peroxiredoxin enzyme overexpression across the total cell population, the transduced cell lines were also assessed by intracellular flow cytometry using a c-MYC monoclonal antibody. High transduction efficiencies of 98%, 100%, and 99% were achieved for the Prx1, Prx2, and Prx6 polyclonal BEL-A cell lines, respectively (Figure 1B). Mean fluorescence intensities (MFI) for c-MYC labelling were normalised to the background levels detected using untransduced (UT) control expanding BEL-A cells to determine a fold-increase in overexpression relative to the background. Prx2 was overexpressed to the highest level (1,200-fold increase), with Prx1 and Prx6 residing at slightly lower levels of 60- and 70-fold increases, respectively (Figure 1C). 

In order to determine the stability of overexpression throughout terminal erythroid differentiation, the expanding Prx1, 2, and 6 BEL-A cell lines were differentiated to generate reticulocytes. By co-labelling day 12 samples with c-MYC antibody and fluorescent Hoechst staining, it was possible to determine expression levels in reticulocytes. Strikingly, flow cytometry data showed that, despite the reduction in cell size during differentiation to reticulocyte, Hoechst negative day 12 reticulocytes (d12r) retained protein expression comparable to that measured on day 0 (Figure 1C). Figure 1D illustrates an enrichment for each of the overexpressed peroxiredoxin proteins observed in the reticulocyte population when compared to a mixed population of day 12 orthochromatic cells and nuclei. These data demonstrate that the expression of each overexpressed Prx enzyme is robustly maintained and retained in reticulocytes with minimal loss during differentiation. 

### 3.2. Overexpression of Glutathione Peroxidase Enzymes in BEL-A Cells

Next, the overexpression of cDNAs corresponding to full-length glutathione peroxidase isoforms GPx1 and GPx4 was explored. Initial expression tests were performed by transducing expanding BEL-A cells with lentiviral vectors comprising the cDNA of full-length primary transcripts for GPx1 and GPx4 (GPx1: ENST00000419783; GPx4: ENST00000354171), incorporating an N-terminal c-MYC tag. However, no expression was detected when assessed by immunoblotting (Figure 2A) or flow cytometry (Figure 2B). The lack of expression of the native cDNA sequences was anticipated, given that GPx proteins contain a selenocysteine (UGA) residue in the catalytic site, which can impede translation if not effectively recoded. In order to circumvent this, the selenocysteine (Sec) residues in each construct were mutated to cysteines (Cys) to assess whether overexpression could be resolved. Although this change from Sec to Cys would likely impact enzymatic activity [75,76], we hypothesised this mutation would allow us to assess whether the expression of GPx is feasible. 

Following lentiviral transduction of BEL-A cells, overexpression of GPx1^U49C^ and GPx4 ^U73C^ was confirmed by immunoblotting using specific antibodies raised against GPx1 and GPx4, respectively, or using a monoclonal c-MYC antibody. A strong single band corresponding to the height of c-MYC-tagged GPx1 was observed in the GPx1^U49C^ BEL-A cell line, with no accompanying alterations to endogenous GPx1 levels observed (Figure 2A). A band corresponding to the predicted molecular weight of full-length c-MYC-tagged GPx4, along with a second smaller band, which was not detected by the c-MYC antibody, was also observed in the immunoblots of the GPx4^U73C^ BEL-A cell line; this is indicative of N-terminal cleavage (Figure 2A). 

Intracellular flow cytometry confirmed that substantial overexpression of c-MYC tagged GPx1^U49C^ was achieved in 99% of the transduced BEL-A cells (Figure 2B). GPx4 overexpression was, however, only partially resolved in the BEL-A cells transduced with c-MYC GPx4^U73C^ cDNA, with only a transduction efficiency of 37% (Figure 2B) and mean 8-fold increase in expression relative to the background c-MYC levels in untransduced (UT) control expanding BEL-A cells (Figure 2C). However, due to the N-terminal positioning of the c-MYC tag for this construct, the occurrence of N-terminal cleavage events may render flow cytometry data measurements under-representative of the total overexpression levels achieved.

In order to assess the stability of overexpression during terminal erythroid differentiation, the expression of GPx1^U49C^ and GPx4^U73C^ was determined by intracellular flow cytometry using a c-MYC monoclonal antibody in day 12 BEL-A reticulocytes (d12r). Around one-quarter of starting GPx1^U49C^ overexpression was lost throughout differentiation (Figure 2C,D). However, due to the high starting levels of overexpression achieved, GPx1^U49C^ BEL-A cell lines retained a substantial abundance of GPx1^U49C^ in reticulocytes (Figure 2C,D). Conversely, GPx4^U73C^ levels were completely lost by day 12 of differentiation (Figure 2C,D). This result was also confirmed by Western blot analysis using a GPx4-specific antibody, whereby no GPx4 protein was detected in the day 12 samples (Figure 2E). Of note, we also observed that endogenous GPx4 protein was difficult to detect in culture-derived reticulocytes via immunoblotting, suggesting that the bulk of this protein is normally lost during erythroid differentiation in humans.

### 3.3. Inhibition of Protein Degradation Pathways 

The limited levels of GPx4 overexpression achieved thus far, in both expanding BEL-A cells and during differentiation, indicated that GPx4 was being regulated/degraded. As ubiquitin-mediated proteasomal degradation and lysosome-induced degradation are considered the two main pathways of erythropoietic protein turnover, the effects of the proteasome inhibitor, MG-132, and lysosomal inhibitor, leupeptin, on GPx4^U73C^ overexpression in expanding BEL-A cells was investigated [77]. Furthermore, since it is also reported that specific HSC70 recognition motifs can trigger GPx4 for degradation via chaperone-mediate autophagy (CMA) [78], we also explored the effect of bardoxolone-mediated CMA inhibition. Expanding GPx4^U73C^ BEL-A cell lines were therefore exposed to each inhibitor or the respective vehicle control for a total of 6 h. The resultant expression levels were determined by intracellular flow cytometry and normalised to background c-MYC labelling in untransduced control BEL-A cells. 

As shown in Figure 2F, an eight-fold increase in c-MYC expression was observed in untreated (NT) expanding BEL-A cells transduced with c-MYC GPx4^U73C^ cDNA. No further increase in c-MYC GPx4^U73C^ expression was observed for any of the vehicle controls (VC) or for c-MYC GPx4^U73C^ transduced cells treated with leupeptin or bardoxolone (Figure 2F). Significant improvement to GPx4^U73C^ overexpression was only observed in c-MYC GPx4^U73C^ transduced BEL-A cells exposed to 5 μM MG-132, whereby inhibition of ubiquitin-mediated degradation resulted in a final 10-fold increase in GPx4^U73C^ overexpression when normalised to background c-MYC labelling in untransduced control BEL-A cells.

### 3.4. Generation of Sec-Containing Glutathione Peroxidase Ubiquitination Mutants

In nature, GPx4 exists in 3 versions: a long form (22 kDa), expressed from the primary transcript predominantly in spermatozoa, a short form (19 kDa), which is generally found in subcellular locations such as the cytoplasm, nucleus, and microsome of other cell types, and a sperm-specific nuclear form, which is translated from an alternative exon located in the first intron of the GPx4 gene [79]. Examination of the full-length cDNA GPx4 sequence used so far in this study revealed a 25-amino-acid mitochondrial leader sequence at the N-terminus. As erythroid mitochondrial proteins are degraded during the later stages of erythropoiesis and during reticulocyte maturation, we hypothesised this signal peptide may result in the degradation of GPx4^U73C^ due to targeting the protein to mitochondria and may account for the cleavage events observed in Figure 2A. The shorter GPx4 transcript, lacking the mitochondrial leader sequence (called sGPx4) was, therefore, taken forward for further study.

Furthermore, as cysteine-containing GPx constructs are expected to be less catalytically active than seleno-containing proteins and more susceptible to inactivation by overoxidation to sulfinic and sulfonic acid forms, establishing a strategy to overexpress a selenocysteine-containing enzyme remained favourable [75,76]. In order to address this, the coding region of the original GPx4 construct was extended to include the native GPx4 3’ untranslated region (UTR) nucleotide sequence (called here GPx4 3’UTR) that contains a Sec insertion sequence (SECIS)—a mRNA structure known to aid the recruitment of tRNA^Sec^ in native conditions [80]. 

Finally, efforts were made to block the observed ubiquitin-mediated GPx4 turnover. As prolonged use of the ubiquitination-mediated degradation inhibitor, MG-132, during cultures is not a viable solution to enhancing GPx4 expression due to toxic effects during differentiation, the GPx4 coding sequence was, instead, examined for putative ubiquitination sites using PhosphoSitePlus [70]. In total, six lysine residues (K80, K99, K121, K135, K140 and K164, as depicted in Figure 3A) were identified as potential targets for ubiquitination in sGPx4. These lysines were identified to be surface-located and, therefore, accessible for ubiquitination. Importantly, an examination of a computational model of the GPx4 structure (based on 7L8Q.pdb) showed that these specific lysines are also not directly associated with the active site, making the mutagenesis of these sites unlikely to impact catalytic activity (Appendix A). All six lysine residues were, therefore, substituted for arginine residues to remove potential ubiquitination sites. This yielded the final construct called sGPx4 3’UTR 6KR (Figure 3A).

BEL-A cells were stably transduced with lentiviral vectors containing cDNA for the following constructs: full-length (FL) GPx4, FL GPx4^U73C^, FL GPx4 3’UTR, sGPx4 3’UTR, and sGPx4 3’UTR 6KR (Figure 3A). Overexpression was determined by intracellular flow cytometry using a c-MYC monoclonal antibody, and total GPx4 expression was confirmed by immunoblotting using a GPx4-specific antibody. As shown in Figure 3B, the inclusion of a 3’UTR sequence following the cDNA region of FL GPx4 was not sufficient to improve expression, with FL GPx4 3’UTR transduced BEL-A cells only exhibiting up to a minimal two-fold increase in expression, as observed previously for the FL GPx4 (no 3’UTR) construct. Interestingly, overexpression levels of sGPx4 3’UTR were similar to that of FL GPx4^U73C^, confirming that the expression of the natural version of a seleno-containing GPx4 can be achieved in erythroid cells using its shorter cDNA form (Figure 3B). Finally, the levels of sGPx4 3’UTR 6KR overexpression achieved were comparable to that of sGPx4 3’UTR in expanding BEL-As, indicating that the lysine mutations introduced were not disruptive to the overall stability of the enzyme (Figure 3B). Immunoblotting using a GPx4-specific antibody also confirmed that the sGPx4 constructs aligned with the cleavage product observed in the FL GPx4^U73C^ samples, confirming that FL GPx4^U73C^ is likely undergoing an N-terminal cleavage event due to the presence of an N-terminal mitochondrial leader sequence (Figure 3C). 

In order to assess the expression of these new GPx4 constructs in day 12 BEL-A reticulocytes, cell lines transduced with sGPx4 3’UTR and sGPx4 3’UTR 6KR were differentiated, and expression levels in reticulocytes were assessed by using intracellular flow cytometry (Figure 3D). As observed previously for GPx4 constructs, there was a substantial reduction in sGPx4 3’UTR construct overexpression, from around nine-fold to three-fold (Figure 3E) between day 0 and day 12 of differentiation. However, the removal of ubiquitination sites resulted in approximately three-fold more sGPx4 3’UTR 6KR in day 12 reticulocytes (d12r) than sGPx4 3’UTR (Figure 3E). This demonstrates that ubiquitination events are actively contributing to the degradation of overexpressed GPx4 during erythroid differentiation and showcases that this loss in expression can be overcome to achieve overall higher levels of GPx4 in culture-derived reticulocytes. 

## 4. Discussion

Reactive oxygen species (ROS) play a critical role in the physiology and pathology of RBCs, influencing cellular functionality and life cycle, contributing to various diseases, and playing a role in the formation of storage lesions. Breakthroughs in our understanding and manipulation of antioxidant systems within these cells may offer significant therapeutic benefits by mitigating oxidative stress and/or enhancing red blood cell longevity and function or storage. Importantly, our work has demonstrated, for the first time, that it is possible to achieve the successful overexpression of antioxidant enzymes in reticulocytes during erythroid culture, and this now reveals an exciting avenue for future antioxidant protein manipulation studies. 

Of the five antioxidant proteins overexpressed in this study, the three peroxiredoxin enzymes were the most straightforward to achieve overexpression during erythroid differentiation. The overexpressed c-MYC tagged Prx1, Prx2, and Prx6 proteins were also observed to be well retained in BEL-A-derived reticulocytes, with Prx2 abundance being enhanced the most without any detectable compensatory change in endogenous protein levels. It is possible that the higher expression of Prx2 compared to the other Prx isoforms reflects the natural capacity of RBCs to express Prx2 at high levels. Interestingly, Prx1 and Prx6 isoforms are reported to be naturally turned over during erythroid terminal differentiation in humans [81,82], yet our study demonstrates that overexpressed cDNA constructs can surpass underlying clearance mechanisms to remain at overall higher levels in culture-derived reticulocytes. For Prx2, perhaps its ability to bind to the N-terminal tail of the highly abundant band 3 membrane protein and/or interactions with haemoglobin may facilitate good reticulocyte retention [33,38]. Given that Prx2 is known to be the third highest expressed protein in red blood cells, it is particularly striking that we can achieve further enhanced levels of expression, illustrating an impressive capacity of erythroid expression machinery for generating more Prx2.

Unlike the peroxiredoxin proteins, the overexpression of glutathione peroxidases proved more challenging. Both GPx1 and GPx4 contain a non-canonical selenocysteine (Sec) residue [80]. Thought to have evolved to enable resistance to overoxidation than cysteine (Cys) residues, Sec incorporation is believed to improve catalytic activity by better preserving enzymatic function [75]. As the overexpression of initial full-length GPx1 and GPx4 constructs was recovered on exchanging Sec for a canonical Cys, it can be concluded that the presence of Sec introduces a level of complexity to the overexpression of GPx1 and GPx4. For GPx4 in this study, the overexpression of a Sec-containing construct was achieved by ensuring the inclusion of the native Sec insertion sequence (SECIS) containing 3’UTR, and we anticipate that a similar manoeuvre will also be effective for GPx1 overexpression. The successful overexpression and retention of GPx4 were also only achieved in BEL-A derived reticulocytes after the removal of its hydrophobic mitochondrial sequence. We hypothesise that the full-length primary transcript of GPx4 (22 kDa) is likely lost due to its mitochondrial localisation, as mitochondria are cleared during reticulocyte maturation. Therefore, the normal localisation or existence of potential cryptic localisation signals should be considered when attempting to overexpress enzymes or other proteins in erythroid cell cytosol to facilitate the production of modified reticulocytes. 

We also demonstrated that overexpressed GPx4 is lost in human erythroblasts during erythropoiesis and that this loss in GPx4 overexpression is sensitive to MG-132, which is consistent with the ubiquitin-mediated degradation of the protein. The abundance of GPx4 in culture-derived reticulocytes was increased by the collective removal of six putative ubiquitination sites, preventing the loss of the protein during erythroid differentiation evoked by ubiquitination events. This reproduces a strategy previously adopted to increase the abundance and retention of thymidine phosphorylase in culture-derived reticulocytes [68]. In the case of GPx4, the mutagenesis of these six putative ubiquitination sites is not expected to hinder function, but future functionality studies are required to confirm this. 

The increase in the recent attention to GPx4 and its possible role in enucleation also motivated our efforts to focus on successfully expressing a selenocysteine-containing GPx4 construct [26,27,28,29]. However, in this study, we did not observe any improvement in reticulocyte yields in our BEL-A culture system. Nonetheless, given the rise in interest in GPx4, the demonstration of how best to achieve GPx4 overexpression in culture-derived reticulocytes provides the field with the means to explore wild-type/mutant GPx4 rescue studies in knockdown/knockout models to help establish the exact role GPx4 plays in erythropoiesis, enucleation, as well as its role in mitophagy; a key step of reticulocyte maturation whereby the mitochondria is cleared from the cell by autophagosomes once reticulocytes have egressed from the bone marrow into circulation [25]. 

In this study, we used the amenability of the BEL-A cell line for genetic manipulation to successfully screen for the overexpression of five antioxidant proteins using lentivirus. We provide data that demonstrate, for the first time, the ability to obtain culture-derived reticulocytes with augmented antioxidant protein expression, highlighting the potential for enhancing oxidative stress defences. The results from this study also present us with antioxidant candidates to take forward for further exploration for use in disease model systems, e.g., β-thalassaemia and sickle cell disease. Future studies using larger-scale production in immortalised cells (although this is currently an expensive option due to the lower enucleation efficiency and higher culture costs) or in primary stem cell cultures would also facilitate the exploration of the potential effects on reticulocyte function and storage and assess how well culture-derived reticulocytes with augmented antioxidant protein expression respond to ROS. It is particularly critical to understand how modified reticulocytes containing enhanced antioxidant proteins behave within the context of the entire RBC antioxidant system so metabolic control analysis (MCA) alongside “omic” studies encompassing metabolomic and lipidomic studies are merited. This is because antioxidant enzymes such as GPx1, GPx4, PRx1, Prx2, and Prx6 do not work alone but rely on complex interactions with other cellular components to remain active and functional. The future use of larger-scale cultures will, therefore, enable us to examine whether the co-expression partner enzymes or supplementation with thiol precursors are required to ensure the catalytic longevity of antioxidant enzymes. The investigation of potential enzymatic interdependencies will provide deeper insights into how best to optimise the overall resilience of erythroid progenitor cells and reticulocytes to oxidative stress.

## 5. Conclusions

Overall, these data demonstrate the feasibility of producing engineered reticulocytes with heightened antioxidant enzyme expression, which have the potential to serve as a multipronged approach to improve the cell’s intrinsic response to oxidative stress and perhaps enhance longevity or storage without the need to administer exogenous antioxidant alternatives. The success of producing therapeutic RBCs ex vivo will depend on being able to efficiently grow cells at scale, administer doses with a suitable half-life, and effectively store large quantities of culture-derived reticulocytes long-term. Manipulating and overexpressing antioxidant enzymes or, indeed, entire metabolic pathways might improve the antioxidant capacity of culture-derived reticulocytes, thereby enhancing their future therapeutic or circulatory potential and/or also enhancing their storage longevity. Thus, the work in this study demonstrates an exciting new strategy to support the pursuit of producing culture-derived therapeutic reticulocytes.

## Figures and Tables

**Figure 1 antioxidants-13-01070-f001:**
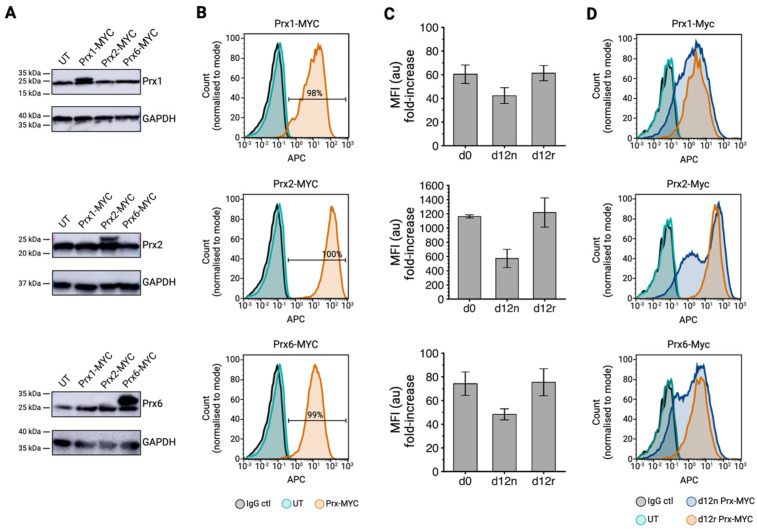
Overexpression of peroxiredoxin enzymes in BEL-A erythroblasts and reticulocytes. (**A**) Representative Western blots of d0 expanding BEL-A cell lysates (Prx1 and Prx6: 1 × 10^6^ cells, Prx2: 2.5 × 10^5^ cells per lane) obtained from indicated cell lines, labelled with monoclonal antibodies against Prx1, Prx2, Prx6, and GAPDH (loading control), *n* = 3. (**B**) Flow cytometry histogram of IgG isotype control (black) and c-MYC (9E10), labelling the fixed and permeabilised BEL-A erythroblasts from untransduced (UT) control cells (turquoise) and the respective Prx 1, 2, and 6 expressing cell lines (orange). (**C**) Bar graphs illustrate the Prx overexpression from indicated cell lines at day 0 (d0) and day 12 (d12) of differentiation. Day 12 samples were gated on Hoechst +/− labelling to obtain data for day 12 nucleated (d12n) cells and day 12 reticulocytes (d12r). Mean fluorescence intensities (MFIs) were normalised to background c-MYC labelling in UT control cells (*n* = 3, mean ± SD). (**D**) Flow cytometry analysis of Prx construct expression in day 12 samples (*n* = 3). Fixed and permeabilised cells were labelled with either an IgG isotype control (black) or c-MYC (9E10) antibody and gated based on Hoechst +/− labelling to obtain data for day 12 nucleated (d12n) cells and day 12 reticulocytes (d12r).

**Figure 2 antioxidants-13-01070-f002:**
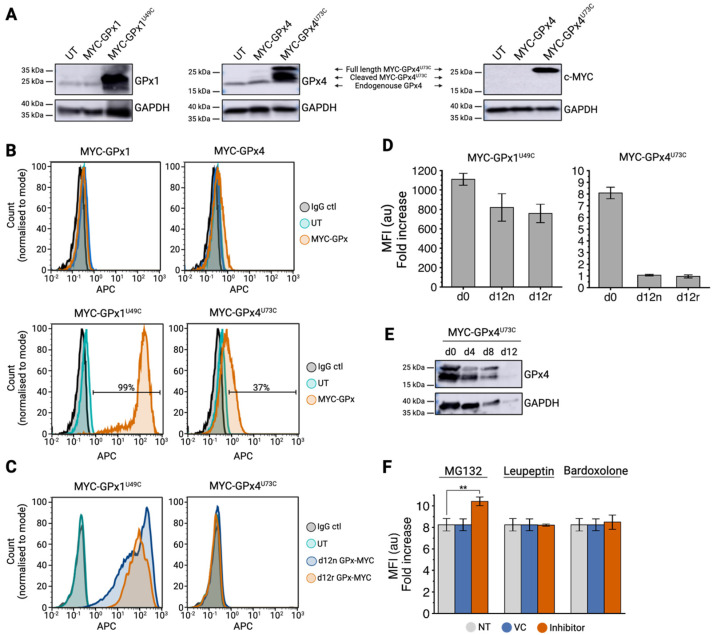
Overexpression of glutathione peroxidase enzymes in BEL-A erythroblasts and reticulocytes. (**A**) Representative Western blots of cell lysates (1 × 10^6^ cells per lane) obtained from the indicated cell lines, labelled with monoclonal antibodies against GPx1, GPx4, c-MYC, and GAPDH (loading control), n = 3. (**B**) Flow cytometry histogram of IgG isotype control (black) and c-MYC (9E10) labelling in fixed and permeabilised BEL-A erythroblasts from untransduced (UT) control cells (turquoise) and the respective GPx transduced cell lines (orange). (**C**) Flow cytometry analysis of GPx construct expression in day 12 samples (n = 3). Fixed and permeabilised cells were labelled with either an IgG isotype control (black) or c-MYC (9E10) antibody and gated based on Hoechst staining to obtain data for day 12 nucleated (d12n) cells and day 12 reticulocytes (d12r). (**D**) Bar graphs illustrating GPx overexpression from indicated cell lines at day 0 (d0) and day 12 (d12) of differentiation. The day 12 samples were gated based on Hoechst staining to obtain data for day 12 nucleated (d12n) cells and day 12 reticulocytes (d12r). Mean fluorescence intensities (MFIs) were normalised to background c-MYC labelling in untransduced control cells (n = 3, mean ± SD). (**E**) A representative Western blot of total GPx4 expression in c-MYC GPx4^U73C^ cell lysates (1 × 10^6^ cells per lane) during differentiation at the time points indicated (n = 2). (**F**) Bar graphs illustrating GPx4^U73C^ overexpression analysed by intracellular flow cytometry following treatment with either 5 μM MG132, 10 μM leupeptin, 10 μM bardoxolone, a vehicle control (blue), or left untreated (NT) (grey) for a total of 6 h. Mean fluorescence intensities (MFIs) were normalised to background c-MYC labelling in UT control cells. Data are shown as the mean ± SD; n = 3; ** *p* < 0.01; two-tailed unpaired Student’s *t*-test.

**Figure 3 antioxidants-13-01070-f003:**
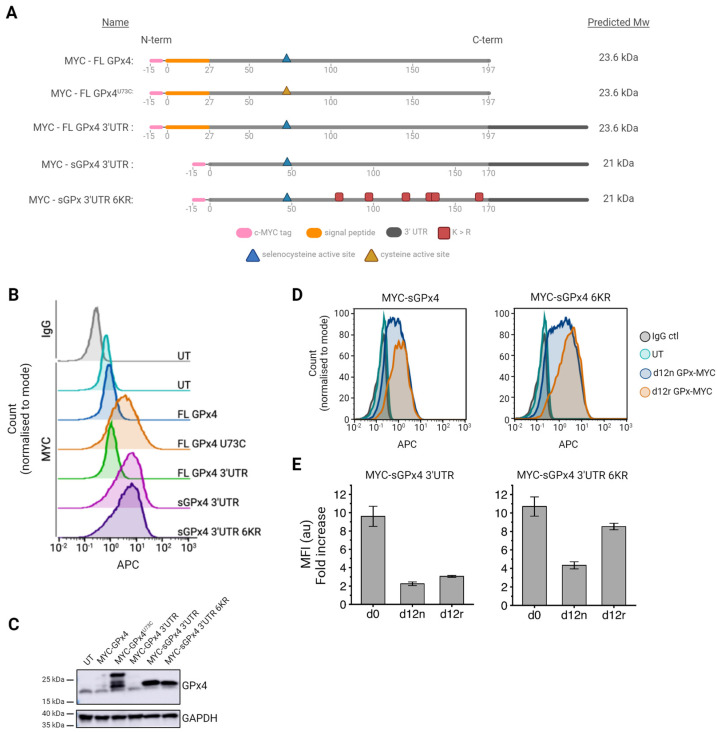
Overexpression of GPx 4 in BEL-A erythroblasts and reticulocytes. (**A**) Schematic of GPx4 constructs tested and their predicted molecular weights. (**B**) Flow cytometry histogram of IgG isotype control (black) and c-MYC (9E10) staining in fixed and permeabilised BEL-A erythroblasts from untransduced (UT) control cells (turquoise) and the respective GPx4 transduced cell lines. (**C**) A representative Western blot of cell lysates (1 × 10^6^ per lane) obtained from the indicated cell lines, labelled with monoclonal antibodies against GPx4 and GAPDH (loading control), n = 3. (**D**) Flow cytometry analysis of GPx4 construct expression in day 12 samples (n = 3). Fixed and permeabilised cells were labelled with either an IgG isotype control (black) or c-MYC (9E10) antibody and gated based on Hoechst staining to obtain data for day 12 nucleated (d12n) cells and day 12 reticulocytes (d12r). (**E**) Bar graphs illustrating GPx overexpression from indicated cell lines at day 0 (d0) and day 12 (d12) of differentiation. Day 12 samples were gated based on Hoechst staining to obtain data for day 12 nucleated (d12n) cells and day 12 reticulocytes (d12r). Mean fluorescence intensities (MFIs) were normalised to background c-MYC labelling in UT control cells (*n* = 3, mean ± SD).

## Data Availability

Data is contained within the article and Appendix A.

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
