# Peer review of "Modulation of Antioxidant Enzyme Expression of In Vitro Culture-Derived Reticulocytes"

_antioxidants, 2024, doi:10.3390/antiox13091070_

Round 1
Reviewer 1 Report
This well-written manuscript describes research that is especially relevant for increasing the length of storage of blood in blood banks. The experiment is well designed and conducted and fulfills its objectives. I have no major suggestions to make.
I have a few specific comments.
line 21: Please place antecedent to this in this sentence to improve intent of the sentence. I suggest that authors go through whole manuscript and add antecedent to this and these when used to begin a sentence.
line 68: Place hyphen between cysteine and dependent.
line 163: Place hyphen between heat and inactivated.
line 522: Delete Please add.
Congratulations on a well-written manuscript.
Author Response
Major comments:
This well-written manuscript describes research that is especially relevant for increasing the length of storage of blood in blood banks. The experiment is well designed and conducted and fulfills its objectives. I have no major suggestions to make.
Response: We thank the reviewer for their very positive feedback of the manuscript and take on board the minor suggested comments – see below.
Detail comments:
I have a few specific comments.
line 21: Please place antecedent to this in this sentence to improve intent of the sentence. I suggest that authors go through whole manuscript and add antecedent to this and these when used to begin a sentence.
Response: We agree and have made this change. We have also applied antecedent throughout the manuscript where we deem appropriate. See P1, L21.
line 68: Place hyphen between cysteine and dependent.
Response: We thank the reviewer for identifying this. We have included a hyphen. See P2, L92.
line 163: Place hyphen between heat and inactivated.
Response: We thank the reviewer for identifying this. We have included a hyphen. See P4, L163.
line 522: Delete Please add.
Response: We thank the reviewer for identifying this. We have deleted the wording in text. See P13, L553.
Congratulations on a well-written manuscript.
Response: Thank you!
Reviewer 2 Report
The manuscript is well-written and the whole project is well-thought and logically structured. I have no major comments. The authors should be praised for their excellent work. Moreover, it's a great addition to both the Special Issue and the selected journal.
- The authors could enrich their Methods section by mentioning the antibodies/inhibitors used, to help the readers better understand the methodological part of the research.
- The authors mention the role of GPX4 in enucleation. GPX4 seems to also play a role in mitophagy (10.3389/fphys.2020.609103), and alterations in its levels might affect this process too. The authors could add this in their well-written and detailed discussion section.
- If possible, it would be interesting to have a graphical visualization of the overall findings, such as the mechanisms (either established or hypothesized) that affect the expression of each studied enzyme (proteasome, mitochondrial retention etc).
Author Response
Major comments:
The manuscript is well-written and the whole project is well-thought and logically structured. I have no major comments. The authors should be praised for their excellent work. Moreover, it's a great addition to both the Special Issue and the selected journal.
Response: We thank the reviewer for their very positive comments.
Detail comments:
- The authors could enrich their Methods section by mentioning the antibodies/inhibitors used, to help the readers better understand the methodological part of the research.
Response: We thank the reviewer for pointing this out. We agree that the antibodies and inhibitors were missing from the methods sections and have added these in. See page P3, L142 and P5, L222, respectively.
- The authors mention the role of GPX4 in enucleation. GPX4 seems to also play a role in mitophagy (10.3389/fphys.2020.609103), and alterations in its levels might affect this process too. The authors could add this in their well-written and detailed discussion section.
Response: We thank the reviewer for this point and for providing the above reference. We agree that GPx4 is also reported to play a role in mitophagy, a key step in peripheral reticulocyte maturation. We have added a line in the discussion to acknowledge the possible contribution that the work we report in this study could make towards studies investigating the role of mitophagy in reticulocyte maturation in health and disease models. See P12, L509.
- If possible, it would be interesting to have a graphical visualization of the overall findings, such as the mechanisms (either established or hypothesized) that affect the expression of each studied enzyme (proteasome, mitochondrial retention etc).
Response: We thank the reviewer for this excellent suggestion. We have produced a summary graphical illustration. This is now the graphical abstraction which has been uploaded.
Reviewer 3 Report
See the attached pdf file
See the attached pdf file

Author Response
Major comments:
General
The authors describe experiments in which they genetically manipulated erythroblasts to over express the antioxidant enzymes peroxiredoxin (Prxs) and glutathione peroxidase (GPxs).
The aim was to decrease the concentration of the free radical species, most notably the superoxide anion that is produced in a ‘side reaction’ between the oxygen bound to heme in oxyhaemoglobin with the concomitant oxidation of the FeII in the heme to FeIII, making methaemoglobin. The reactions typically leads to this oxidation in 1-2% of the oxyhaemoglobin in a red blood cell (RBC) per day. The re-reduction occurs via methaemoglobin reductase (diaphorase) this ensuring long term function of the RBC as an oxygen transporter.
The authors explain in the Introduction that there are several antioxidant enzyme systems in RBCs acting in parallel. They give no indication of the relative fluxes through these pathways and do not quantify concentrations of even the main free radical. They do not appreciate whether simply increasing the concentration of the two key enzymes necessarily enhances the rate of ‘detoxification’ of the free radicals or to what extent other reaction ‘step up’ and carry out the reaction(s) if one or other of the enzyme is defective or deficient. In other words, in many pathways the flux control coefficient of an enzyme can be quite low and altering its concentration has little effect on overall pathway flux. In relation to this article, the sort of considerations made with metabolic control analysis (MCA) are missing.
Response: We thank the reviewer for this insightful comment. Whilst we recognise that flux analysis would be an interesting contribution to this work going forward, we feel that it remains outside of the scope of this study. The aim of this research was to investigate whether the overexpression of antioxidant enzymes could be achieved in erythroid cells given they already have a significant contingent of antioxidant enzymes. Essentially is there extra capacity? We would also like to highlight the various papers described and referenced in the introduction that illustrate how increasing these enzymes does have the potential to enhance “detoxification” in erythroid cells (see P3, L124). In our discussion, we recognise that these enzymes are components of multiple different complex pathways, the control and harnessing of which will require further work (see P12, L528). As such, we have proposed that future ‘omics’ studies should be conducted to help elucidate the wider cellular effects overexpressing these enzymes might have but agree that this discussion point could be extended to include flux analysis in helping to identify which pathway(s) might be the most promising to focus on. We have added this point in here P12, L525.
Major criticisms/comments
The Introduction starts out in a general way but careful parsing of the opening sentences reveals “woolly” or imprecise thinking. I can illustrate this as follows.
- Page (P) 1 Line (L) 28 “…at risk of oxidative stress mediated damage due to their high iron content and oxygen transport role”
Let me parse this: oxidative stress mediated damage…the “oxidative stress” (meaning potential for oxidation) does not mediate the “damage” …what mediates the damage is electrons transferred from chemical moieties to acceptor atoms including oxygen…this electron transfer “mediates” the reaction and the oxidation brings about the damage that alters the conformation of structural proteins, enzymes, membrane transport proteins, and membrane lipids, rendering them inactive or interacting in different ways that alters their function(s). The “risk” is an imprecise way of describing redox potential. The claim of “high iron content” is imprecise. These is little free iron (Fe2+ and Fe3+) in erythrocytes; it is molecularly “tied up” in the heme of haemoglobin where it has a high oxygen affinity. This iron is central to the oxygen carrying capacity of the haemoglobin. So, the “iron content” and this physiological function of “oxygen transport” are effectively the same thing.
- P1 L29 “Over time, the spontaneous autooxidation of haemoglobin (Hb) results in production of highly reactive oxygen species (ROS) that elicit…” Again parsing: “Over time”…is some sort of delay meant here? Or are you implying that there is a cumulative effect (hence “over time”) of the many different oxidation reactions? Specifically in relation to oxyhaemoglobin, there is a continual ‘side reaction’ between oxygen that is bound to FeII in heme of oxyhaemoglobin and the majority of the oxygen which is “off-loaded” to tissues; this ‘side reaction’ oxidizes the FeII to FeIII making methaemoglobin, and a superoxide radical molecule. So, the statement above is incorrect. It is not “autooxidation of haemoglobin” (a seemingly catchy term) but oxidation of FeII in oxyhaemoglobin with the formation of the oxygen radical species, superoxide. This is commonly called (as in this article) “reactive oxygen species (ROS)”. This acronym has become an emotive term that obscures proper science. In a dispassionate scientific article on the oxidation of FeII in haemoglobin (which is well understood), it is more appropriate and more precise to refer to superoxide. Approximately 1-2% of the heme in haemoglobin is oxidized to methaemoglobin per day. So, perhaps unsurprisingly there has evolved the methaemoglobin reductase pathway that reduces the FeIII to FeII. This is a well characterized pathway
- P1 L38 “….continual exposure to partial oxygen conditions…” does not make sense. Do you mean ‘low oxygen partial pressure p[O2]’? Low p[O2] is known to promote the polymerization of sickle cell haemoglobin. Does this “trigger” “autoxidation”? The altered sickle cell haemoglobin structure may well have increased reactivity to form methaemoglobin. Why not state this much more precise description of the events?
- P1 L39. It is claimed that the elevation of [ROS] leads to inactivation of Ca-ATPase and activation of (implied) mechanosensitive cation channel(s) (Piezo1). But the distortion of the erythrocyte shape by ‘sickling’ alone activates Piezo1 thus elevating cytoplasmic [Ca2+]…without the “need” to postulate a decline in Ca-ATPase. The latter is well known to have a maximal activity ~3x greater than Na,K-ATPase in the human erythrocytes.
Response to comments 1-4:We have taken on board the reviewers comments in the above points. We appreciate that the introduction may have been slightly generalised in terms of our descriptions and appreciate the suggestions. We have amended the introduction to be more specific and have also shortened it as much as possible.See P1, L29.
- P2 L44. “Dysregulated ROS…”. Again, this is imprecise/fuzzy expression. Perhaps explain what is “regulated” about the handling of superoxide in normal erythrocytes and what aspects of this become “dysregulated” in, say, sickle cells.
Response: We think the use of dysregulation is suitable as it has been used many times in this situation to refer to disease states and where altered ROS levels occur. However, to simplify we have changed the wording to just say “ROS”. See P1, L43. We wish to keep dysregulation where mentioned in the abstract.
- P2 L57. “…self perpetuating ROS…” is pure hype (unscientific)…what is meant by this statement?
Response: Thank you for highlighting this. To simplify this sentence we have removed this wording from the text. See P2, L53.
- P2 L65 “…three enzyme systems acting in concert…”. The latter term implies some form of ‘coordination’ between these pathways (under the control of a ‘conductor’); but this is dispassionate chemistry in action; and the reactions simply occur in parallel at rates that are determined by the ‘enzyme kinetics’ (concentration dependence of the various enzyme kinetic parameters including inhibition constants)
Response: We believe “concert” is correct to use in terms of English language as it suggests together or jointly (we are not implying a literal concert with a conductor!). But to help clarify this sentence we have changed the wording from “concert” to “parallel”. See P2, L61.
- P2 L68. “…consumption of GSH…”. The moiety is not “consumed” per se, but merely oxidized to the disulfide compound GSSG, which is recycled back to GSH via glutathione reductase.
Response: As the reviewer suggests, to help clarify this sentence we have changed the wording from “consumed” to “oxidised” to better describe the mechanism of action. See P2, L62.
- P2 L71 “non-canonical”, “catalytic tetrad”, “paramount”…Explain what you mean without the “glitz”
Response: We believe “non-canonical” and “catalytic tetrad” are appropriate acceptable scientific terms. “Non-canonical” is a scientific definition used in biology to describe amino acids that are not encoded by the genetic code, including selenocysteine. The term “catalytic tetrad” simply refers to the 4 amino acids required for catalytic function. We have extended this sentence to describe all 4 said amino acids. See page P2, L66. We have changed the wording of “paramount” to “essential” to describe the necessity of the active site residues on GPx1 function and now reads “…is essential for GPx1 neutralisation of H2O2”. See P2, L62.
- “…iron-dependent oxidised lipids…”. I am not aware of any lipids that are iron dependent. What is meant here. Perhaps you referring to the Fenton reaction?
Response: Thank you for this point. To help clarify this sentence we have changed the wording of this sentence to: “… a type of cell death dependent on iron that is characterised by lipid peroxidation.” We believe this to be a clearer definition of ferroptosis. See P2, L73.
- P2 L92. The point about a possible ‘moonlighting’ action of glutathione peroxidase as a structural protein in erythroblast enucleation is an important (novel) hypothesis; and the first (to me) novel idea in this Introduction
Response: We thank the reviewer for this positive comment.
- P3 L101 “…protecting RBCs against ROS”. A much less hyped (and more scientific/precise/chemical) way of stating this is: ‘catalysing the dismutation of superoxide anion in parallel with the abovementioned enzymes.’
Response: We kindly disagree that this sentence is “hyped” as it is not just superoxide we need to protect the cells from, the cells suffer exposure of different ROS. As such, we prefer to remain more general here in our description and have not changed it.
- P3 L105. What is meant by “the resolving cysteine”. This term needs an explanation or is it simply a distraction from the main message of the sentence and could be left out?
Response: We believe the term “resolving cysteine” to be appropriate. The term is correct from the literature and refers to the second catalytic cysteine in peroxiredoxins that is required for enzymatic activity. We believe this sentence to therefore be an accurate description of how the enzyme functions. See P3, L99.
- P3 L122. Why not simply state ‘storage related oxidation’ and leave out the “stress” term.
Response: To help simplify this sentence we have removed this wording from the text and changed the sentence to: “storage related oxidation’”. See P3 L110 and L114.
- The Materials and Methods sections are comprehensive
Response: We thank the reviewer for this positive comment.
- P6 L280…this claim is not justified (in the absence of hard data in your hands; although claimed by a Ref 83 and 84 in another cell system)
Response: We thank the reviewer for this comment. We have adjusted this to say “Although this change from Sec to Cys would likely impact enzymatic activity [75,76], we hypothesized this mutation would allow us to assess whether the expression of GPx is feasible”
- The Results section is long and rambling. It needs to be logically developed using division into sections with well chosen
Response: We agree that headings could help to better divide and guide the reader. As such, we have sectioned the results under appropriate headings. See P5, L239; P7, L289; P5, L331; P9, L373; P5. The results section has been kept as concise as possible and this was not highlighted as an issue by the other reviewers.
- P9 L366…what is meant by “hyperoxidation”? How does this differ from being ‘less stable’ due to altered tertiary structure etc?
Response: To help clarify this sentence we have changed the wording of this sentence to “more susceptible to inactivation by over-oxidation to sulfinic and sulfonic acid forms”. We believe this to better describe the mechanism of inactivation. See P9, L386.
- P9 L371…here is a section that could easily have been introduced under a heading like‘ Suppression of ubiquitin effect’…and “computational modelling”, where is this described in Materials and Methods?
Response: We thank reviewer for pointing out this example. As addressed for point 17, we have included headings throughout the results section. For this specific example, see P6, L373.
The mention of computational modelling on P9, L401, was referring to the production of a computational model of GPx4 protein structure using standard methods based on a available structure in the pdb database. To address this point we have added more information about how this was done in the methods section (See P5, L231) and we have moved the person involved to middle authorship as they wrote the methods and provided the references. We still refer to the actual model as data not shown as the structure was not affected so we simply state this. We can provide an image of this model if it is required.
Minor criticisms
- P4 L153 use SI superscript convention “cells/mL” -> ‘cell mL^-1’…etc throughout the article
Response: We thank the reviewer for this edit. We have amended these units throughout the manuscript. See P4, L153.
- P4 L157 “hours” -> ‘h’
Response: We thank the reviewer for this edit. We have amended these units throughout the manuscript. See P4, L158.
- P4 L191, “media” -> ‘medium’
Response: We thank the reviewer for this edit. We have amended this term throughout the manuscript. See P4, L157.
- P4 L198 use proper multiply sign…here and throughout the article (e.g., Fig. 1 caption)
Response: We thank the reviewer for this edit. We have amended this throughout the article. See P4, L153.
- P5 L235 “antioxidant over expression” -> ‘antioxidant-enzyme over expression’
Response: We thank the reviewer for this edit. We have included the word “enzyme” here and in any other similar cases throughout the article. See P6, L252.
- P5 L247 “enucleated reticulocytes” -> ‘reticulocytes’ as by definition reticulocytes do not have nuclei so “enucleated is a redundant adjective…see also P6 L254
Response: We thank the reviewer for this edit. We have removed the word ‘enucleated’. See P6, L264.
- P7 L312 “indicated to us” -> ‘suggested’…”us” is implied…”in some way” of course it would be “in some way” but this is hardly a scientific explanation…modify the sentence
Response:We thank the reviewer for this edit. We have removed uncertain language and modified the sentence to: “indicated that GPx4 was being regulated/degraded”. See P7, L333.
- P11 L435 “periredoxins” -> ‘periredoxin’
Response: We thank the reviewer for identifying this. We have made ‘peroxiredoxin’ singular. See P11, L460.
- P11 L442 What do you mean by “natural capacity”? Do you mean ‘maximal catalytic capacity’ or better still ‘high flux control coefficient’?
Response: We would like to kindly point out that we are not referring to the activity of the enzyme here but expression. To clarify this we have edited the text to read: “reflects the natural capacity of red blood cells to express Prx2 at high levels”. See P11, L466.
- P12 L487 “cultured reticulocytes’ -> a better term since reticulocytes like erythrocytes lack a nucleus means they can’t be “cultured” in the usual sense of the word would be ‘culture derived reticulocytes’…use here and elsewhere where the term is used.
Response: We thank the reviewer for this point. For better clarity, we have modified the text to “culture-derived reticulocytes” here and throughout the manuscript. See P11, L469.
- P12 L506 insert ‘enzyme’ before “expression”
Response: We thank the reviewer for identifying this. We have included the word “enzyme”. See P12, L536.
- P12 L510 …what has “economic viability” go to do with this science?
Response: We have taken on board the reviewer comments and we have altered the text to simplify our point. It now reads: “The success of producing therapeutic RBCs ex vivo, will depend on being able to efficiently grow cells at scale, administer doses with a suitable half-life, and effectively store large quantities of culture-derived reticulocytes long-term.” See P12, L539.
- P12 512…”lab-grown RBCs” is nonsensical, or at least not a natural extrapolation of the ideas emerging from the work described in this article
Response: We thank the reviewer for this comment. Lab grown blood is a more general accessible term used in the field. We have modified the text to “culture-derived therapeutic reticulocytes” as a better extrapolation from this work. See P13, L546.
Overall
This article has an Introduction that is substantial too long for what is needed to set the context for the work. It is over hyped and scientifically imprecise in many of the aspects that are described. The Introduction could be reduced in length by 70% and use made of heading s to provide focus on the key concepts. The Materials and Methods section is very detailed and yet it lacks a section on data analysis and quantification, or the statistical assessment of the data; in other words n = values for the key experiments. The Results section is very long and lacking in a logical structure. It also could be reduced by a factor of two and use made of well-chosen headings to help guide the reader. The Discussion should be divided under headings that relate directly to the Results headings. The Conclusions are “soft” and nonquantitative.
Responses:We thank the reviewer for their interest in our study and for providing their suggestions. We believe we have addressed the majority of these. For specific changes made please refer to the points above.
We have included alterations to the introduction to reduce it in length. We have not cut it by 70% as it's important to ensure there is adequate narration and background explanation for the different antioxidant enzymes expressed. It's also important to explain what has been done before in this area. Edits were made to better clarify scientific descriptions. We believe the quantification; n numbers and statistics of data analysis is adequately described in the text and figure legends and does not require additional explanation.
Overall, we have shortened the manuscript where we deem appropriate and taking on board the reviewers comments wherever possible. We believe the length of the results to be necessary as we are taking the reader through multiple sequence changes and then describing the results. To address this point, we have included headings to help better section this work and guide the reader and we thank the reviewer for this suggestion. We prefer to leave the introduction and discussion without headings, as to not disturb the flow and we highlight that the two other reviewers were happy with how the manuscript was written.
We disagree with describing this manuscript as “over hyped and scientifically imprecise”. This was not commented on by the other reviewers. They complimented us on the well written article. We looked to make our writing interesting to read and accessible and this may have impacted scientific descriptions for the reviewer but we believe the alterations that the reviewer suggested has improved this. Thank you for this.
Round 2
Reviewer 3 Report
The authors have responded positively to the various comments. [But the other Reviewer must have been looking for different writing attributes if they claimed (as stated by the authors) that the original version was "well written"]. I was well aware of the large amount of work that had gone into the experiments that are reported in the manuscript; so I spent a huge amount of effort on trying to convey the subtleties that needed addressing. The manuscript is much better now and science is the winner. However, I remain unconvinced that the authors see why it is unlikely that simply boosting the concentration (activity) of key enzymes (that are already very active) that are involved in detoxifying ROSs will affect red blood cell functionality and enhance survival . On the other hand, that can be addressed in subsequent work; at least the molecular technology for enhancing the activities of the selected enzymes has been developed and sufficiently well described for others to repeat/test the work.
L22 'culture-derived'
Author Response
The authors have responded positively to the various comments. [But the other Reviewer must have been looking for different writing attributes if they claimed (as stated by the authors) that the original version was "well written"]. I was well aware of the large amount of work that had gone into the experiments that are reported in the manuscript; so I spent a huge amount of effort on trying to convey the subtleties that needed addressing. The manuscript is much better now and science is the winner. However, I remain unconvinced that the authors see why it is unlikely that simply boosting the concentration (activity) of key enzymes (that are already very active) that are involved in detoxifying ROSs will affect red blood cell functionality and enhance survival . On the other hand, that can be addressed in subsequent work; at least the molecular technology for enhancing the activities of the selected enzymes has been developed and sufficiently well described for others to repeat/test the work.
Response: OK thank you.
Detail comments
L22 'culture-derived'
Response: We have now. altered this. Thank you to the reviewer for all your input in improving the manuscript.